# A Multi-Featured Factor Analysis and Dynamic Window Rectification Method for Remaining Useful Life Prognosis of Rolling Bearings

**DOI:** 10.3390/e25111539

**Published:** 2023-11-13

**Authors:** Cheng Peng, Yuanyuan Zhao, Changyun Li, Zhaohui Tang, Weihua Gui

**Affiliations:** 1School of Computer, Hunan University of Technology, Zhuzhou 412007, China; pengcheng@hut.edu.cn (C.P.); yyzhao@hut.edu.cn (Y.Z.); 2School of Automation, Central South University, Changsha 410083, China; zhtan@csu.edu.cn (Z.T.); whgui@csu.edu.cn (W.G.)

**Keywords:** rolling bearings, health indicator construction, stochastic fluctuations, remaining useful life, degradation model

## Abstract

Currently, the research on the predictions of remaining useful life (RUL) of rotating machinery mainly focuses on the process of health indicator (HI) construction and the determination of the first prediction time (FPT). In complex industrial environments, the influence of environmental factors such as noise may affect the accuracy of RUL predictions. Accurately estimating the remaining useful life of bearings plays a vital role in reducing costly unscheduled maintenance and increasing machine reliability. To overcome these problems, a health indicator construction and prediction method based on multi-featured factor analysis are proposed. Compared with the existing methods, the advantages of this method are the use of factor analysis, to mine hidden common factors from multiple features, and the construction of health indicators based on the maximization of variance contribution after rotation. A dynamic window rectification method is designed to reduce and weaken the stochastic fluctuations in the health indicators. The first prediction time was determined by the cumulative gradient change in the trajectory of the HI. A regression-based adaptive prediction model is used to learn the evolutionary trend of the HI and estimate the RUL of the bearings. The experimental results of two publicly available bearing datasets show the advantages of the method.

## 1. Introduction

Rolling bearings are one of the most common rotating machines in industry, and their performance directly affects the health of the entire machine or piece of equipment. Mechanical failures caused by bearing failures often lead to casualties, plant shutdowns, economic losses and even catastrophic accidents if maintenance measures are not taken in a timely manner [1,2,3]. Accurate remaining life predictions help to better develop maintenance strategies, thus improving the reliability of machinery and equipment [4,5,6]. Therefore, the remaining life prediction of bearings has received a lot of attention from researchers.

Overall, a remaining useful life prediction is performed as part of condition-based maintenance and is designed to predict the remaining useful life of a machine, based on historical and ongoing degradation trends observed from condition monitoring information. The remaining useful life prediction is usually divided into four main steps: data collection, health indicator construction, health stage classification and remaining life prediction [7,8]. Monitoring data, such as vibration signals, are first collected from the sensors. Then, appropriate health indicators are constructed to mark the current health status, using methods such as signal processing techniques or artificial intelligence techniques. Different stages of degradation are classified according to the health indicators. Finally, a physical model or neural network is used to estimate the remaining useful life of the machinery. Current research in remaining life predictions focuses on constructing appropriate health indicators, determining the first prediction time, determining failure thresholds and mitigating fluctuations in health indicators [9,10,11,12].

Studies have found that the accuracy of health indicators will affect the accuracy of remaining life prediction, and constructing appropriate health indicators can help improve the predictability of mechanical equipment. Health indicators can be divided into two categories: physical health indicators and virtual health indicators. Briefly, physical health indicators are usually extracted from vibration signals using statistical methods. In contrast, virtual health indicators are mainly constructed by fusing multiple physical health indicators. Various physical characteristics are used as health indicators in many studies, such as root mean square (RMS) and peak values. Based on the instantaneous definition of Shannon’s spectral entropy, Civera et al. [13] used the instantaneous spectral entropy as a health indicator, thereby achieving condition monitoring of wind turbines. In fact, due to the complex internal structure and variable operating environment, a single physical feature extracted from raw monitoring data cannot fully capture failure trends over the lifetime of a machine [14,15]. In feature level fusion studies [15,16,17], by building an optimized model of features, the optimal features are selected for constructing health indicators. However, there are many random fluctuations in the health indicators constructed based on multi-feature fusion, and there is no solution for such stochastic fluctuations in the above method. Ahmad et al. [18] have proposed a linear correction technique to deal with random fluctuations, to fit the increasing trend of health indicators. In Yan et al. [19], a smoothing method is proposed, based on different regression models, to deal with anomalous fluctuations in health indicators. The above methods ensure the health indicators have a clear monotonic trend by processing the stochastic fluctuations, which satisfies the prognostic requirement of the monotonicity of the health indicators. Regarding the effective handling of spurious fluctuations [20], it would be beneficial to identify the starting point of bearing degradation, which in turn would improve the prediction accuracy. However, these methods weaken the degradation trend of the bearings during processing and cannot accurately distinguish the different degradation stages of the bearings.

In addition, determining the first prediction time is a worthy concern [21,22], which will affect the accuracy of the RUL estimate. The root mean square (RMS) is the most commonly used physical property that reflects the increase in vibrational energy with degradation [23]. Figure 1 shows the complete process of a rolling bearing degrading from a healthy state to degradation to failure, using RMS as a health indicator. In the healthy stage, the curve shows a relatively smooth trend along with stochastic fluctuations; after a failure occurs, the curve first shows a linear trend and is in the slow degradation stage; as the damage continues to increase, the bearing enters the accelerated degradation stage. If the prediction takes into account the starting point of bearing degradation, it is impossible to accurately calculate the duration of the bearing in the slow degradation stage and the accelerated degradation stage, respectively, and it is therefore difficult to estimate the remaining life of the bearing. Elbow point detection is based on the premise that there are two apparently different degradation trends, slow degradation and accelerated degradation. The bearing degradation trend at the onset of degradation is linear, while it becomes closer to exponential during the accelerated degradation phase as the failure deepens and spreads. Rigamonti et al. [24] proposed an elbow point identification method based on a Z-test. However, the features describing the trend of bearing degradation do not satisfy the prerequisites of the Z-test well. Baptista et al. [25], using different neural network models to detect elbow points, incorrectly identified spurious fluctuations as elbow points during detection, due to the presence of spurious fluctuations in the actual degradation process of the bearing.

There is a proliferation of methods for determining remaining useful life predictions, but some issues remain a focus of attention in this area:(1)Owing to the complex internal structure and variable external environment, none of the features can fully capture all the degradation information. Therefore, it is difficult to attain effective and accurate results using a single degradation feature as a health indicator.(2)Considering the differences between individual bearings and the uncertainty of failure propagation, the duration of different degradation stages in the full life cycle possesses a certain degree of randomness. In some existing studies, the first prediction time is mostly distributed in the early stage of degradation. The uncertainty of the duration of the slow degradation stage reduces the accuracy of the prediction results.(3)Generally, the state of degradation of rolling bearings is monotonic. This means that once a failure has occurred, the damage it causes is irreversible. However, stochastic fluctuations carry disturbing information for the process of predicting the RUL. An effective solution to the problem of how to deal with anomalies caused by spurious fluctuations is urgently needed.

To solve the above problems, this paper proposes a health indicator construction and prediction method based on multi-featured factor analysis. The method processes the original vibration signal using wavelet denoising. Based on the consideration of information entropy, the relationship between the signal and noise is balanced by adjusting the threshold value, thus changing the signal-to-noise ratio to maximize the extraction of useful information. First, the extracted features are factor analyzed to mine the hidden common factors. The variance contribution ratio is calculated with the goal of maximizing the characterization of the bearing degradation trend, thus completing the construction of health indicators. After considering the interference of anomalies caused by spurious fluctuations on the first prediction time, the cumulative gradient change within the sliding window is analyzed for elbow point detection, i.e., the first prediction time is determined. In view of the fact that stochastic fluctuation is not a degradation feature of the bearing, we design a dynamic window rectification method to deal with it, which ensures the monotonicity of the bearing. Finally, the degradation trajectory of the bearing is learned based on an adaptive degradation regression model, thus accomplishing the prediction of the remaining useful life of the bearing.

The rest of the paper is organized as follows: Section 2 describes the methodology of this paper, including feature extraction, health indicator construction, dynamic window correction, elbow detection and RUL prediction. Section 3 includes experiments and discussions. Section 4 concludes the whole paper.

## 2. Methodology

In this section, a new statistical method is proposed for the RUL estimation of rolling bearings. The whole prediction process is shown in Figure 2. The overall method can be divided into five parts: feature extraction, health indicator construction, elbow point detection, dynamic window correction and RUL prediction.

### 2.1. Feature Extraction

This section focuses on how to extract features from a large amount of historical monitoring data using signal processing techniques. Noise interference is unavoidable during signal extraction [6]. In this paper, the original data are decomposed by the basic wavelet threshold denoising method. After the threshold quantization of the high-frequency coefficients of each layer of the wavelet decomposition, an estimate of the wavelet coefficients is obtained, and then the signal is reconstructed with the inverse wavelet transform to obtain the denoised vibration signal. In the study, Daubechies wavelets were chosen, in order to be as similar as possible to the denoised signal. Using a wavelet denoising method in theory should balance the relationship between signal and noise by adjusting thresholds, thereby changing the signal-to-noise ratio to maximize the extraction of useful information. Since the choice of threshold value may lead to overfocus, i.e., when the amplitude of the wavelet coefficients of the noise decreases with the increase in the decomposition scale, the threshold value is fixed to 0.2 after comprehensive analysis. 

Time domain analysis characterizes the health of mechanical equipment by calculating the time domain statistical features of the vibration signal. To a certain extent, the time domain features can reflect the complete process of performance degradation of mechanical equipment, but it is not sensitive to the occurrence of initial failure. Among them, the more widely used time domain indicators include 12 features, such as root mean square, peak to peak, impulse and crest. The calculation of the 12 features is given in Table 1.

### 2.2. Health Indicator Construction

Since overlapping information among features increases the computational effort and the difficulty of analyzing the problem, there is a necessity to construct health indicators that sufficiently contain degradation information and have little redundancy and a low dimensionality. Factor analysis starts from inter-feature correlation, explores the basic structure in the data by studying the internal dependencies between multiple features, and uses a few hidden variables to represent the basic structure of the original data. The hidden variables can reflect the main information in the original data, and hence such hidden variables are called factors.

Before conducting factor analysis on the extracted features, it is necessary to perform a test of adequacy. The Kaiser–Meyer–Olkin (KMO) test is used to assess whether the data are suitable for factor analysis. First, a correlation matrix is calculated, and then the KMO is calculated using the following Formula (1). Here, $s_{ij}$ represents the elements in the correlation matrix, and $e_{ij}$ represents the corresponding residual term. The range of the KMO value is between zero and one. Generally, data with KMO values less than 0.5 are considered unsuitable for factor analysis. The closer the KMO statistic is to one, the stronger the correlation between the variables and the weaker the partial correlation, indicating better results for factor analysis.
(1)$$KMO=\frac{\sum s_{ij}^{2}}{\sum s_{ij}^{2}+\sum e_{ij}^{2}}$$

Let $X_{1},X_{2},X_{3},\ldots ,X_{p}$ be *p* features after normalization, and assume that the features $X_{i}\left(1\leq i\leq p\right)$ satisfy:(2)$$X_{i}=a_{i1}F_{1}+\cdots +a_{im}F_{m}+\varepsilon_{i}$$
and satisfy:
(1)m≤p;(2)$Cov\left(F,\varepsilon \right)=0$;(3)$D\left(F\right)=diag\left(\lambda_{1},\lambda_{2},\ldots ,\lambda_{p}\right)$;(4)$D\left(\varepsilon \right)=diag\left(\sigma_{1}^{2},\sigma_{2}^{2},\ldots ,\sigma_{p}^{2}\right)$.
Among them, $F={\left(F_{1},F_{2},\ldots ,F_{m}\right)}^{T}$ are mutually independent hidden common factors, $A_{p\times m}\left(a_{ij}\right)$ is the factor loading matrix, $a_{ij}$ denotes the loading of the feature $X_{i}$ on the factor $f_{j}$, and $\varepsilon ={\left(\;\varepsilon_{1},\varepsilon_{2},\ldots ,\varepsilon_{m}\right)}^{T}$ are the random error terms.

Let $\lambda_{1}\geq \lambda_{2}\geq \ldots \geq \lambda_{p}$ be the eigenvalues of the correlation coefficient matrix R, and $\eta_{1},\eta_{2},\ldots ,\eta_{p}$ be the corresponding standard normalized eigenvectors. The loading matrix A of the correlation coefficient matrix R after factor analysis is:(3)$$A=(\sqrt{\lambda_{1}}\eta_{1},\sqrt{\lambda_{2}}\eta_{2},\ldots ,\sqrt{\lambda_{m}}\eta_{m})$$

The factor loading $a_{ij}$ is the correlation coefficient between feature $X_{i}$ and common factor $F_{j}$. When the absolute value of the loading of a feature in the common factor is larger, it indicates that the common factor is better suited to characterizing the feature. To solve the correlation coefficient matrix R of the original features, the number of factors is selected by combining the eigenvalues greater than one and the cumulative contribution of variance. Since the factor loadings are not unique, the common factor obtained by factor rotation is more meaningful in practice. The results of such an analysis reduce the subjectivity of interpretation and are often more acceptable than those of principal component analysis.
(4)$$S_{j}=\overset{m}{\underset{i=1}{\sum }}a_{ij}^{2}$$

After calculating the variance contribution rate $S_{j}$ of the rotated common factor $f_{j}$, the common factor with the largest contribution rate is chosen as the characterization of bearing degradation trend HI, and the HI range is limited to [0,1], assuming that the HI corresponding to the failure state is one.

### 2.3. Elbow Point Detection

In order to properly observe the potential relationship between the degradation behavior of the signal and the decreasing RUL, the starting point of the beginning of the accelerated degradation phase needs to be detected, i.e., the elbow point detection. Ideally, a bearing in the normal operating stage will have an HI value that remains essentially constant. Once the bearing starts to degrade, the HI value will continue to increase. Due to the influence of the external environment and the propagation of the bearing’s own damage, some spurious fluctuations usually occur even during the normal operating stage. Studies have shown that these spurious fluctuations can affect the detection of the accelerated degradation point of the bearing, thus affecting the accuracy of a RUL prediction. In order to avoid the spurious fluctuations on the detection of the accelerated degradation point, a linear regression model is fitted on the sliding window of the health indicator, and the continuous change in the gradient of the linear regression model is used to determine the accelerated degradation point of the bearing. The parameters *k* and *b* are determined by the least squares method as shown in the following equation. The expressions for parameter *k* and parameter *b* are given in Equations (6) and (7):(5)y=k×x+b
(6)$$k=\frac{\sum x_{i}y_{i}-\frac{\sum x_{i}\sum y_{i}}{n^{\prime }}}{\sum x_{i}^{2}-\frac{k\sum x_{i}}{n^{\prime }}}$$
(7)$$b=\frac{\sum y_{i}-k\sum x_{i}}{n^{\prime }}$$
(8)$$arg_{k,b}min\left\{\overset{n^{\prime }}{\underset{i=1}{\sum }}{\left(y_{i}-kx_{i}-b\right)}^{2}\right\}$$

The parameter *k* indicates the gradient of the health indicator with respect to time. The elbow point appears between the slow degradation stage and the accelerated degradation stage, implying that the gradient of HI is highly variable at this moment. The spurious fluctuations caused by the abnormal point can also cause abrupt gradient changes, and the starting point of degradation cannot be accurately detected based only on the gradient reaching a fixed threshold. Therefore, this paper proposes that once the gradient corresponding to the samples in the window increases continuously and exceeds the threshold *m* times, the bearing is considered to have entered the accelerated degradation stage at the current moment, and the moment is the starting point of accelerated degradation. In the experimental stage, based on the stability and accuracy of the algorithm, the gradient threshold of the bearing is set to 0.001, and the number of consecutive times *m* exceeded is set to five times. This setting is considered that the abnormal state caused by random noise does not persist in the normal operation stage.

Consideration of the size of the sliding window is essential. In the paper, we analyzed the length of sliding windows on two public datasets [26,27]. According to different window lengths, health indicators are divided into multiple subsequences. The conditional entropy of the subsequence is calculated as the information loss under the current sliding window length. The length of the sliding window is determined to be between 30 and 50 by calculating the information loss and the rule of thumb. Figure 3 illustrate the impact of different window sizes on elbow detection. On the IMS bearing dataset, the corresponding FPTs are 8010 s, 6780 s and 6530 s when the window sizes are 30, 40 and 50, respectively. Similarly, the corresponding FPTs on the PHM2012 bearing dataset are 7160 s, 7220 s and 4980 s, respectively. As can be seen from the two detail figures, the performances of the window sizes at 30 or 50 can both lead to bias in the determination of the FPT, due to differences in individual bearings. The results show that, when the sliding window size is 40, the degradation point of T2-B1 is 6780 s and the degradation point of bearing2-2 is 7220 s. From the figure, it can be seen that the elbow point detection method proposed in this paper can accurately identify the accelerated degradation points and effectively avoid the limitation of spurious fluctuations.

### 2.4. Dynamic Window Rectification

Considering the influence of spurious fluctuations of the bearing during the normal operation stage on the identification of degradation points, this paper therefore proposes a sliding window based gradient continuous change approach to determine the accelerated degradation points. Since the stochastic fluctuations caused by failures are not a degradation characteristic of the bearing, modeling the stochastic fluctuations after the degradation point in the health indicator is impractical from a modeling point of view. Therefore, this paper proposes a dynamic window correction method for dealing with stochastic fluctuations after degradation points. The dynamic window correction method consists of two steps: the determination of the window size and the stochastic fluctuation correction criterion. The degradation of the bearing found during the experiment is irreversible. Once a fault occurs, the health of the bearing does not improve over time. At the same time, the degradation process of bearings is gradual and does not change abruptly during operation. Over a short period of time, the degradation trend of the bearing remains essentially the same or changes slightly.

The sliding window was found to take values in the range 30–70. The size of the sliding window at different times depends on the sample variance within the current window. Firstly, $present_{var}$ is initialized to infinity and it is assumed that the current window size is 30. When $current_{var}\leq present_{var}$, the current window size is kept unchanged; conversely, the window size is increased in increments, and $present_{var}$ is updated. During the window size increase (to a limit of 70), if $current_{var}\leq present_{var}$ is satisfied, the current window size is returned; conversely, the window size is decreased in decrements. After determining the window size at the current moment, the mean and standard deviation of the samples, corresponding to the current window, are calculated and the specific adjustment is shown in the following equation:(9)$$\mu =\frac{1}{n}\overset{n}{\underset{i=1}{\sum }}win_{HI\left(i\right)}$$
(10)$$\sigma =\sqrt{\frac{1}{n}\overset{n}{\underset{i=1}{\sum }}{\left[win_{HI\left(i\right)}-\mu \right]}^{2}}$$
(11)$$win_{HI\left(i\right)}=\left\{\begin{array}{cc} \mu & win_{HI\left(i\right)}>\frac{\max\left(win_{HI\left(i\right)}\right)+\min\left(win_{HI\left(i\right)}\right)}{2}\\ \mu -3\sigma & \;\;win_{HI\left(i\right)}<\min\left(win_{HI\left(i\right)}\right)\\ win_{HI\left(i\right)} & else \end{array}\right.$$

When $\;win_{HI\left(i\right)}>\frac{\max\left(win_{HI\left(i\right)}\right)+\min\left(win_{HI\left(i\right)}\right)}{2}$, $\;win_{HI\left(i\right)}$ is adjusted to the mean value for the samples in the current window; when $win_{HI\left(i\right)}<\min\left(win_{HI\left(i\right)}\right)$, $\;win_{HI\left(i\right)}$ is adjusted to the lower bound of the 3*σ* interval; other sample points that do not satisfy the above conditions are taken unchanged. As shown in Figure 4, the experiment shows that the HI_original, adjusted in the above way, obviously shows a monotonic trend, which eliminates the influence of random fluctuations on the prediction accuracy well.

### 2.5. RUL Prediction

In this study, the HI will grow exponentially once the bearing enters the accelerated degradation stage after a failure occurs. Therefore, once the accelerated degradation point of the bearing is detected, the proposed degradation model is used to predict RUL. To fit the known samples after the degradation point, and then simulate the degradation trajectory of HI reaching the failure state, the failure time corresponding to the HI failure is obtained, and RUL can be expressed as the product of the time difference between the failure time, the current time and the step size of the adjacent time difference. In Formula (12), r,s,g are adjustable linear parameters determined by least squares, $HI_{pre}$ is the predicted HI, T is the running time, $T_{1}$ is the failure time, $T_{0}$ is the current time and Δt is the step size of the neighboring time difference:(12)$$HI_{pre}=re^{sT}+gT$$
(13)$$RUL_{\;}=\left(T_{1}-T_{0}\right)\times \Delta t$$

### 2.6. Prediction Model Construction Process

The flow of the method proposed in this paper is described step by step in the following:(1)First, the raw vibration signals acquired on the sensors are processed. The signal is decomposed into wavelet coefficients at different scales using a wavelet threshold denoising method. Considering the statistical characteristics of the vibration signal and the noise level, the threshold value is determined at 0.2. After reconstruction by inverse wavelet transform, the denoising effect is evaluated using the signal-to-noise ratio. If the information entropy of the denoised signal is close to the information entropy of the signal itself, it means that the denoising method is able to effectively remove the noise and retain the important information of the signal.(2)Time domain statistical features in the signal are extracted for subsequent factor analysis using traditional methods.(3)The normalized features are fed into the factor analysis model to obtain mutually independent hidden common factors. In order to better interpret the factors, orthogonal rotation of the factors is required. Then, the factor loading matrix is analyzed, to determine the strength of the relationship between each feature and the common factor. The variance contribution rate of each common factor is calculated, and the common factor with the largest variance contribution rate is selected as the health indicator, characterizing the bearing degradation trend.(4)The main purpose of elbow point detection is to find the first prediction time. A linear regression model is fitted on the sliding window of the health indicators, and the cumulative gradient change is observed, to determine the location of the first prediction time. The method proposed in this paper indicates that, when the gradient of the samples in the window increases continuously and exceeds the threshold value of 0.001 five times or more, the current moment is considered to be the first prediction time.(5)The HI value of the samples within the current window are analyzed in the case of iterative updating of the window size. Then, adjustments are made according to the given formula so that the HI satisfies the characteristic of a monotonic trend of the bearing during degradation.(6)After determining the first prediction time of the bearing and eliminating the random fluctuations, the known sample data after this degradation point are fitted to simulate the trajectory of the health indicator when it reaches the failure state, so as to calculate the time difference between the failure state and the current state, and to obtain the predicted results.

## 3. Experiment and Result Analysis

### 3.1. Experimental Platform and Dataset

In this study, experimental data were from the IEEE PHM2012 Data Challenge [28] and the University of Cincinnati Intelligent Maintenance Systems (IMS) Center bearing dataset. The applicability of the method was demonstrated with two run-to-failure datasets.

The data of the IEEE PHM2012 Data Challenge were collected on the PRONOSTIA platform, as shown in Figure 5. Table 2 provides a detailed description of the composition of the dataset. The platform was designed by the French FEMTO-ST Institute and consists mainly of a rotating part, a degradation generating part and a measuring part. It is able to provide realistic experimental data for describing the degradation of rolling bearings during their entire life. The platform is able to provide constant and variable operating conditions to accelerate the degradation of the bearing, while collecting monitoring data (speed, temperature, vibration, etc.) in real time. The experiments considered data from three different loads (i.e., 1800 rpm and 4000 N, 1650 rpm and 4200 N and 1500 rpm and 5000 N) and collected vibration accelerations in both horizontal and vertical directions of the bearing housing, with a sampling frequency of 25.6 kHz and 2560 samples recorded every 10 s. For safety reasons, once the amplitude of the vibration signal exceeded 20 g, the test was stopped, and the bearing was considered to have failed. The entire data set contains all possible types of bearing failures during the degradation process.

The IMS bearing dataset was obtained from the NASA Ames Prognostics Data Repository. Table 3 provides a detailed description of the composition of the dataset. These data contain three datasets, each containing full life cycle vibration data for four bearings. For each bearing in dataset 1, one accelerometer was fitted in the x and y axes, respectively, and for dataset 2 and dataset 3, one accelerometer was fitted in each bearing. The sampling frequency was 20 kHz and the duration of each sample was 1 s. A data file containing 20,480 sampling points was generated. The data file was named after the time of data collection, and each record was a data point.

### 3.2. Health Indicator Construction

Twelve statistical features are extracted from the raw data using signal processing techniques, requiring an adequacy test before factor analysis can be performed. Bartlett’s spherical test and KMO test are used to show that this type of data can be used for factor analysis. The correlation matrix of the population variables is examined to see if it is a unit matrix, and if it is not a unit matrix, it indicates that there is a correlation between the original variables and factor analysis can be performed. Correlations and skewness correlations between variables are tested and take values between zero and one. The closer the KMO statistic is to one, the stronger the correlation between variables and the weaker the skewness correlation, therefore the better the factor analysis is. In this paper, the IMS bearing dataset was used as an example for illustration. A factor analysis model was developed for the extracted 12 features, the factors were rotated using variance maximization and three common factors were selected after factor analysis. To observe more intuitively which features were more relevant for each hidden variable, a heatmap of the factor matrix was drawn, based on the absolute values of the correlation coefficients. Figure 6 shows the relationship between different bearings and the common factors, from which it can be seen that the correlation between the three common factors was weak and the data dimensionality reduction was achieved. Then, the variance contribution rate of the rotated common factors was calculated, to select the common factor with the largest contribution rate as the health indicator of the bearing. Figure 7 shows the variance contribution rate for different bearings. Figure 8 shows the health indicators constructed based on multi-feature factor analysis.

Evaluation criteria for health indicators can quantitatively assess the performance degradation curve properties of health indicators. Common criteria [29] for assessing health indicators include correlation, monotonicity and discreteness. To verify the effectiveness of the proposed method, the proposed method-based health indicators are compared with those based on traditional methods and deep learning networks [30]. This paper has been validated on the PHM2012 bearing dataset. The results are shown in Table 4. The results show that the monotonicity of the health indicators, based on the proposed method, is improved, the correlation is essentially the same, and the discreteness is significantly reduced. This shows that the method of factor analysis model is effective. The common factor with the largest contribution to the variance can preserve information while showing good monotonicity as a health indicator. The method presented in this paper takes into account the statistical properties of the vibrational signal of a single bearing, such that the discreteness is small. Health indicators constructed based on machine learning methods take into account the historical monitoring data of different bearings at the same time. The constructed health indicators have a large scatter, due to individual differences. The lower the discreteness of the health indicator, the more accurate the prediction of the remaining useful life based on the degraded trajectory will be.

### 3.3. Elbow Point Detection

The proposed elbow detection method is used to determine the first prediction time, which improves the accuracy of the RUL prediction. Figure 9 and Table 5 reflect the results of the proposed method, in comparison with other methods. Since the duration of the slow degradation phase is difficult to estimate, detecting the accelerated degradation point is more helpful to improve the prediction accuracy. In Ref. [5], the calculated relative root mean square (RRMS) was used as a health indicator to determine whether the bearing had entered the degradation stage, using the most recent *n* samples of the RRMS. In Ref. [28], 3σ criteria were introduced for the RMS as a health indicator, the upper bound was used as an abnormality detection criterion, a degradation state was considered to have been reached when a number of consecutive values were outside this threshold. The energies of the wavelet packet coefficients were extracted in Ref. [31] to characterize the health of the machine, and then the first prediction time was determined based on the proposed multivariate statistical process control (MSPC) method. In contrast to this paper, other methods cannot clearly distinguish between slow and accelerated degradation points and are not well suited to avoid the effects of spurious fluctuations.

### 3.4. Dynamic Window Rectification

As stated in Section 2, stochastic fluctuations due to faults are not degenerate features of the bearing and therefore it is not reasonable to model them. After detecting the accelerated degradation point of the bearing, the stochastic fluctuations in the health indicator are corrected using the proposed method. Figure 10 shows that the proposed method works well to eliminate stochastic fluctuations that do not belong to the bearing degradation trend, and it can be used to estimate the RUL of a bearing after meeting the prognosis criteria of the health indicator. The experiment shows that the corrected health indicators can predict the failure point in advance and reduce costs and losses.

### 3.5. RUL Prediction

After constructing the health indicators, the proposed elbow point detection algorithm is executed to find the accelerated degradation points of the bearing, which is validated using the IEEE PHM2012 challenge data. Figure 11 depicts the prediction process after the degradation point of bearing3. According to the proposed method, the degradation point of the bearing is detected at the time 13,310 s, and then the RULs at 15,000 s, 15,200 s and 15,400 s after the degradation point is predicted. The RULs at the corresponding moments, based on the degradation model, are predicted to be 2200 s, 2270 s and 2280 s, respectively. The experimental results show that the degradation trajectory fitted by the proposed method converges better to the actual RUL as the monitoring time progresses. In practice, this is consistent with our requirement for RUL prediction, that is, more accurate prediction results are expected when the bearing is close to the failure state. This result can be explained by the fact that, at the early stage of failure, the HI values are not sufficient to accurately estimate the parameters of the model, and the more HI values are obtained, the more accurately the parameters of the model are estimated, and the bias that exists at this time is mainly due to the random error brought by the stochastic process. In this study, both the smoothing procedure of HI and the linear regression procedure are able to reduce the stochastic error to some extent. Thus, as more data are collected on the sensor, the RUL prediction is closer to the true value.

### 3.6. Discussion and Comparison

In this study, two commonly used performance metrics, the cumulative relative accuracy (CRA) and the convergence metric, as proposed in the literature [29], were calculated, in order to illustrate the superiority of the method in a comparative manner. The CRA reflects the sum of the relative prediction accuracy of the forecasting methods at a given time and is used to assess the prognostic performance of different methods. The convergence metric reflects how quickly the prediction method approaches the actual RUL during the prediction process. Table 6 summarizes the results of the comparison between the proposed method and the other two methods, which were validated on the PHM2012 bearing dataset. It is worth noting that these methods are based on the prediction of the constructed HI after determining the PFT. The results show that the proposed method has higher CRA scores and better convergence between the three models for all tested bearings. Compared to the other two methods, the proposed method has more accurate prediction results and performs best in the bearing prediction process. 

In addition, the mean absolute error (MAE) [29] and the normalized RMS error (NRMSE) of the predictions in the PHM2012 dataset are calculated separately. Among them, RMS is a commonly used traditional health indicator in the field of residual life prediction, principal component analysis (PCA) is a machine learning method for constructing virtual health indicators, and extreme learning machine auto code (ELM-AE) is a deep learning method. These four existing methods for constructing health indicators, and the proposed method, are used to validate the effectiveness of predicting the remaining service life of bearings based on health indicators. The evaluation criteria are used to analyze it, as well as the results, are shown in Figure 12. Compared with a single structured deep learning network, the proposed method can effectively increase the prediction accuracy. At the same time, it improves the accuracy of the remaining useful life prediction, compared to traditional health indicator construction methods.

These results show that the health indicators constructed according to the method proposed in this paper are consistent with the bearing degradation trend. This is because of the irreversibility of the mechanical degradation process, i.e., it is impossible to recover from the damage once a failure has occurred without artificial repair. The health indicators constructed based on this paper have the inherent property of monotonicity, which meets the prognostic criteria. Meanwhile, the accuracy of the prediction is improved after the elbow point is determined, which means that the first prediction time is determined accurately and can accurately classify the slow degradation and accelerated degradation stages. The remaining useful life prediction, by combining it with the adaptive regression model pointed out in this paper, is in line with the degradation trajectory of the bearings, thus achieving better prediction results.

## 4. Conclusions

In this work, we propose a multi-featured factor analysis-based construction of health indicators that effectively addresses stochastic fluctuations in health indicators, and also propose a novel elbow-point detection method for abnormal fluctuations in the normal operation phase. A factor analysis model is built on the original features, and health indicators are constructed based on the maximization of the factor variance contribution rate. The degradation model is used to predict the RUL at different times, by correcting the stochastic fluctuations after the degradation point, through a dynamic window. Experimental results show that the health indicators constructed in this paper can better characterize the degradation trend of bearings. Based on the correct detection of the bearing degradation points, the dynamic window rectification method successfully handles the stochastic fluctuations in the degradation process, which not only makes the health indicator monotonic, but also better preserves the degradation trend of the bearing, thus improving the prediction’s accuracy. 

Nevertheless, the method proposed in this paper has some shortcomings. First, the method is applicable to degradation processes with slow and accelerated degradation stages but is not applicable to the case of near-sudden bearing failure. Secondly, the accuracy of the prediction will be affected by the construction of health indicators. If the health indicators contain more information that is not part of the bearing degradation, it is impractical to model this part. Maintenance decisions based on inaccurate health indicators may lead to wrong actions, such as unnecessary or delayed maintenance. Finally, the methodology in this research was validated on two publicly available datasets, and the generalizability of the methodology needs to be further evaluated on additional datasets. In future works, we would like to take into account the multi-stage nature of bearing degradation. The process of predicting the remaining useful life of bearings is realized by learning the degradation trend of bearings, in combination with deep learning models for the characteristics of different degradation stages. The construction of health indicators also requires multidimensional considerations, such as temporal and spatial dimensions. Although the application scenarios included in the public dataset are representative, considering the complexity of real industrial environments, more bearing operation-to-failure test data will be explored.

## Figures and Tables

**Figure 1 entropy-25-01539-f001:**
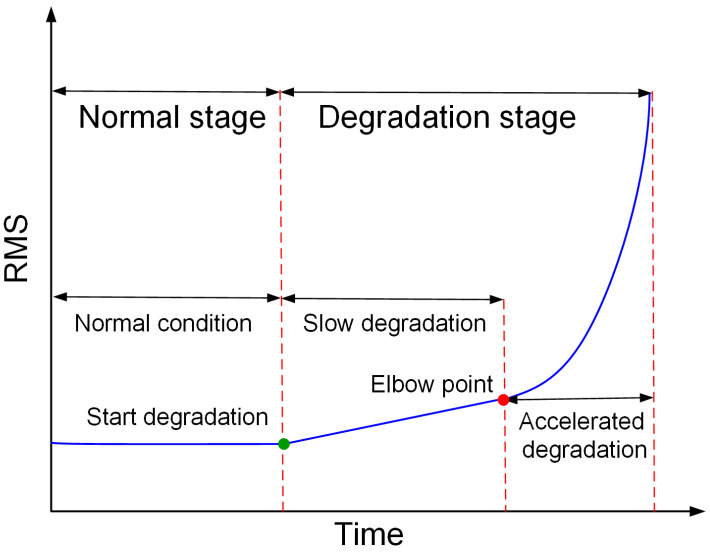
Bearing degradation trajectory.

**Figure 2 entropy-25-01539-f002:**
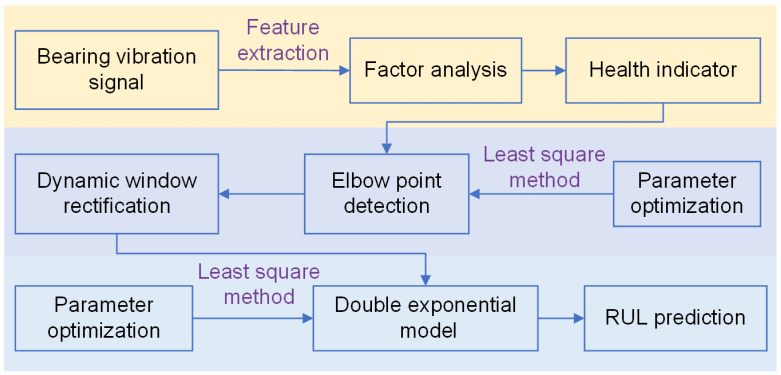
Framework of prediction method in this paper.

**Figure 3 entropy-25-01539-f003:**
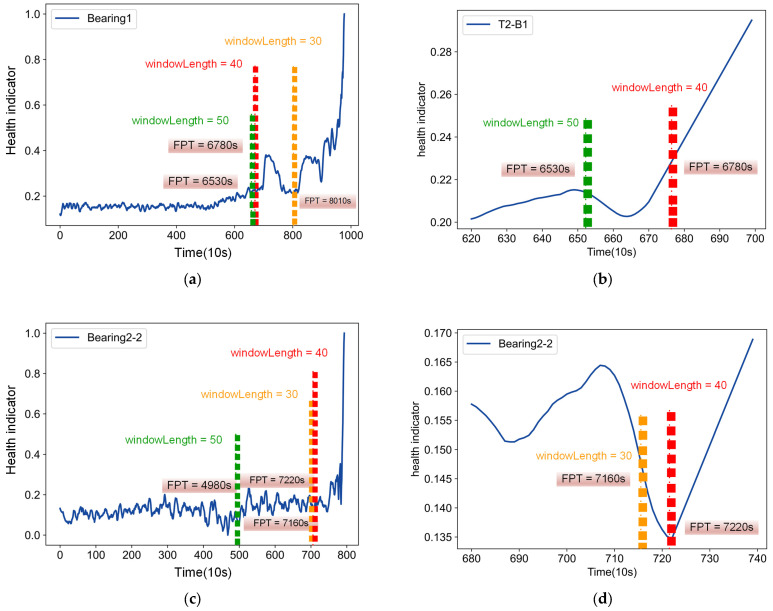
Performance of the proposed method on different bearing datasets. (**a**) IMS bearing dataset T2-B1. (**b**) Detailed drawing of T2-B1. (**c**) PHM 2012 bearing dataset of Bearing2-2. (**d**) Detailed drawing of Bearing2-2.

**Figure 4 entropy-25-01539-f004:**
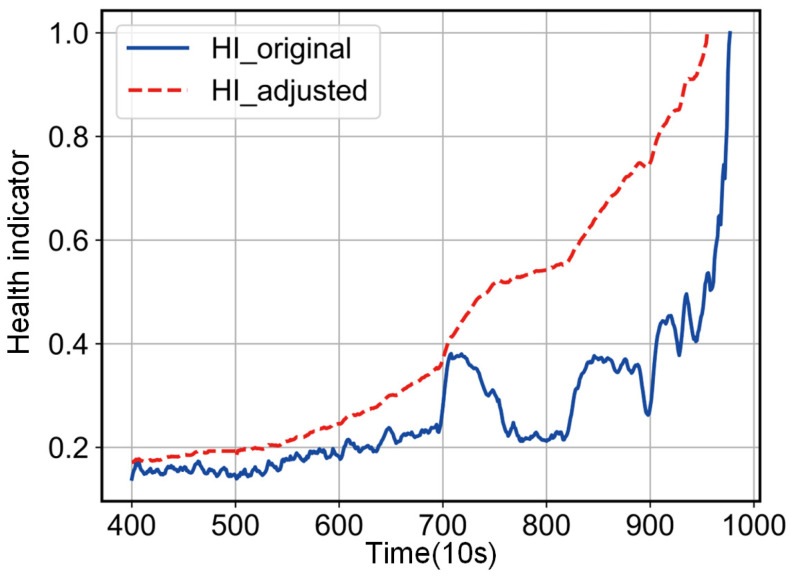
Dynamic window rectification results.

**Figure 5 entropy-25-01539-f005:**
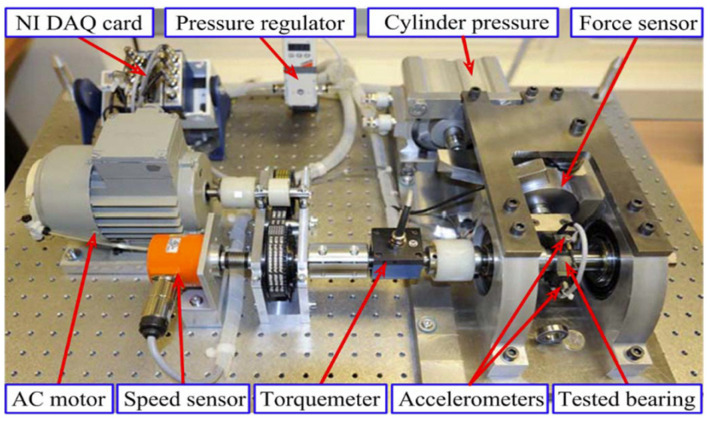
Introduction to the experimental platform.

**Figure 6 entropy-25-01539-f006:**
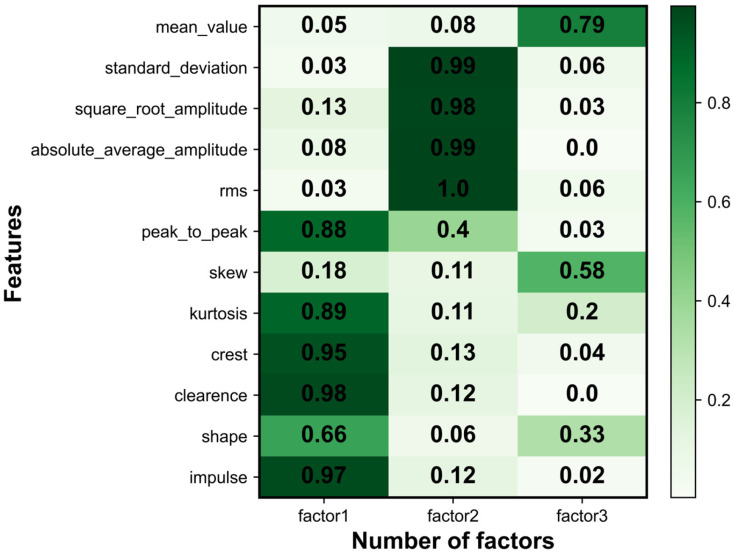
The heat map of feature and factor relationship.

**Figure 7 entropy-25-01539-f007:**
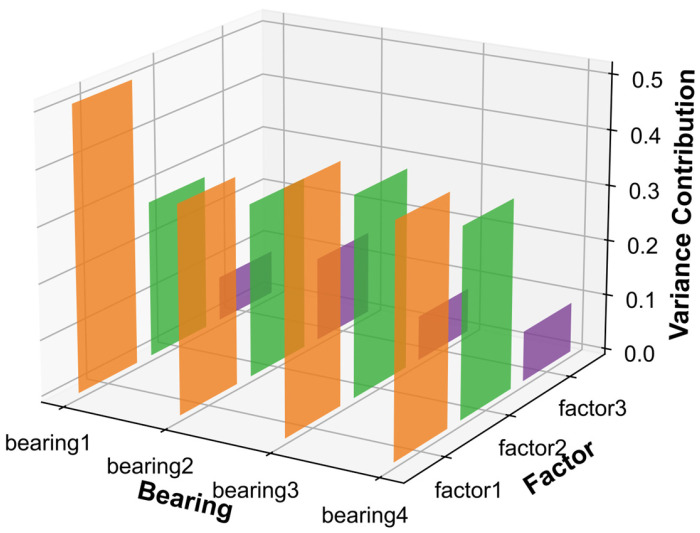
The variance contribution of the common factor.

**Figure 8 entropy-25-01539-f008:**
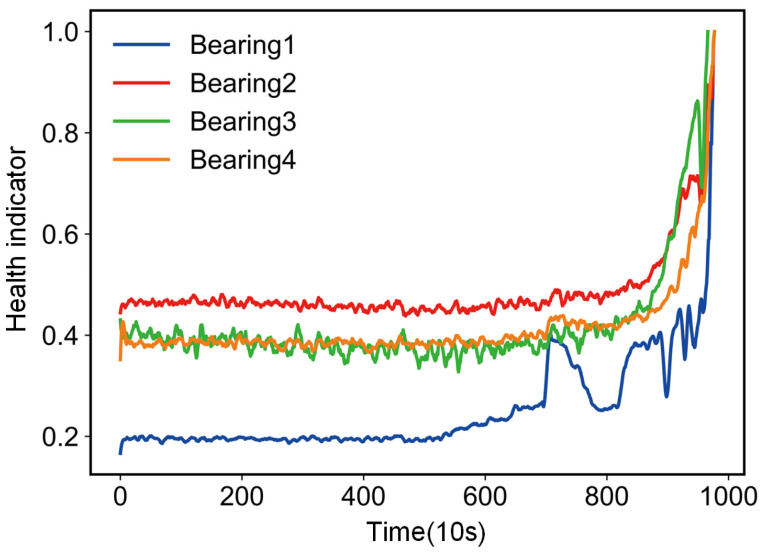
Health indicators for different bearings.

**Figure 9 entropy-25-01539-f009:**
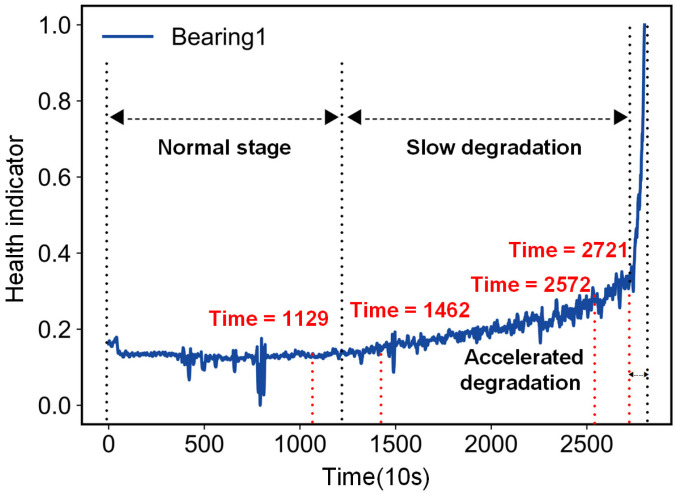
Bearing1-1 elbow point detection.

**Figure 10 entropy-25-01539-f010:**
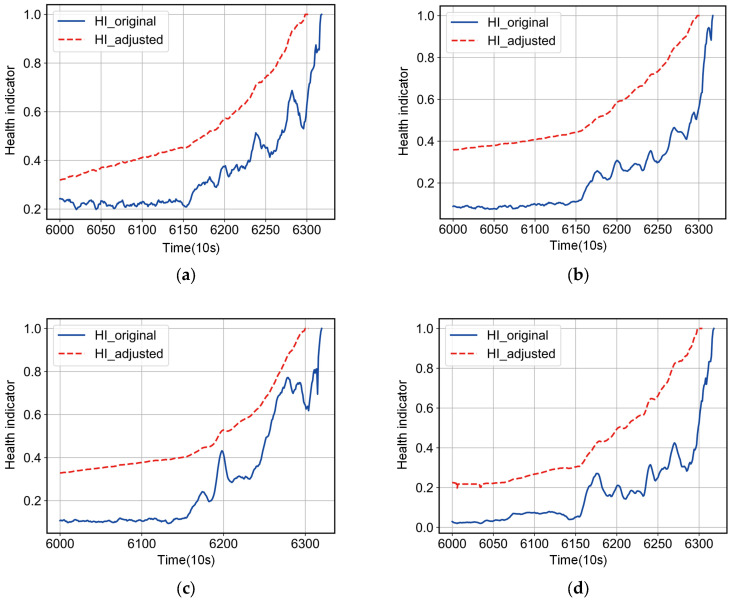
IMS bearing dataset correction results. (**a**) Bearing1. (**b**) Bearing2. (**c**) Bearing3. (**d**) Bearing4.

**Figure 11 entropy-25-01539-f011:**
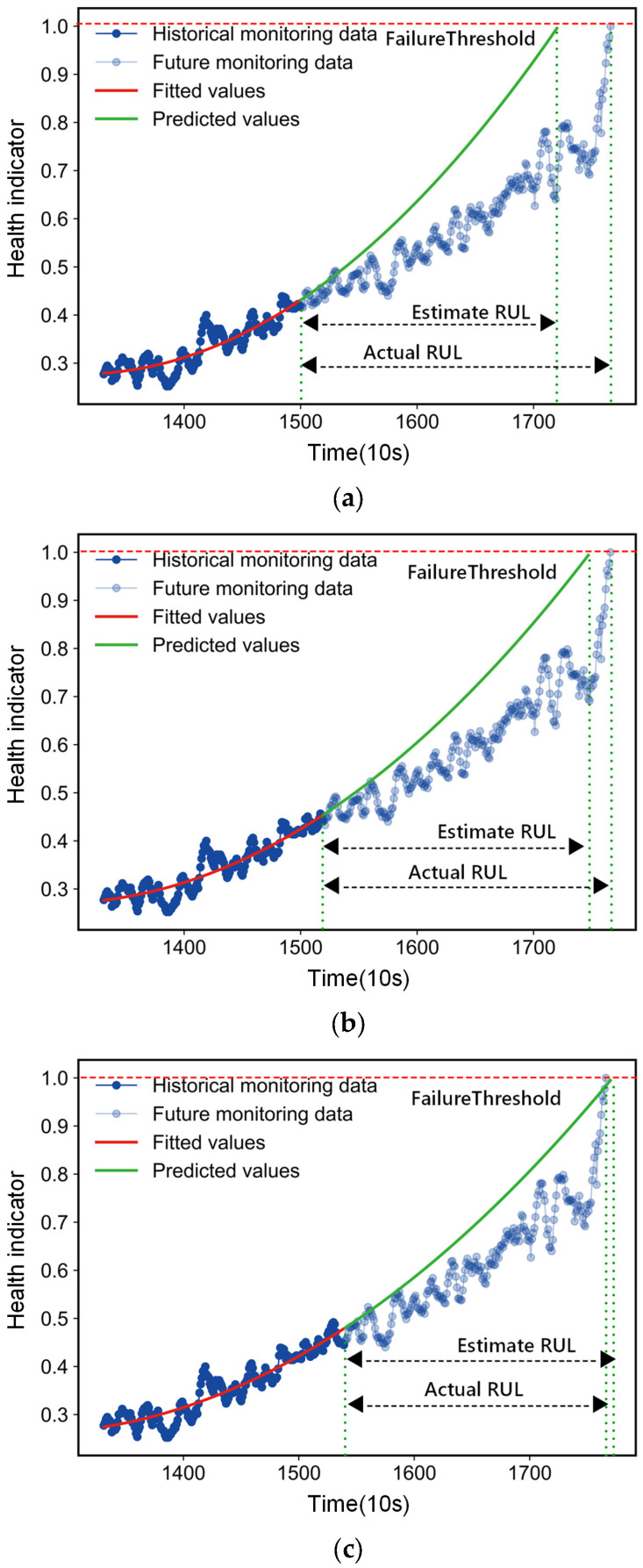
RUL prediction process of bearing3. (**a**) RUL predicted trajectory at the time 15,000 s. (**b**) RUL predicted trajectory at the 15,200 s. (**c**) RUL predicted trajectory at the time 15,200 s.

**Figure 12 entropy-25-01539-f012:**
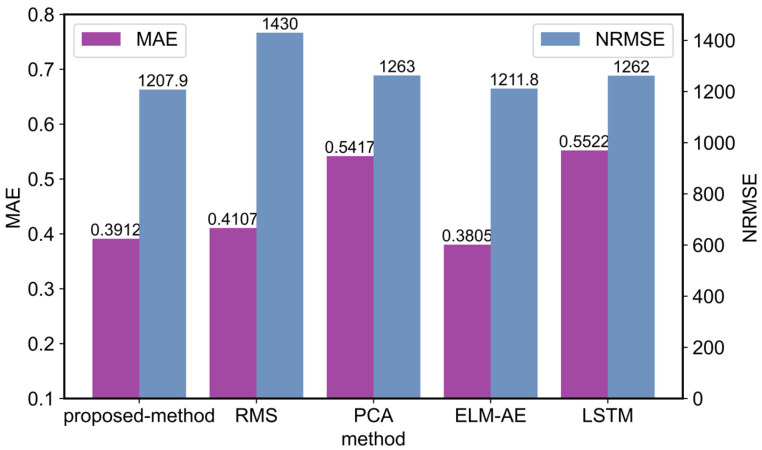
Comparison with other methods.

**Table 1 entropy-25-01539-t001:** Common statistical characteristics.

Features	Equation	Features	Equation
mean	$\bar{X}=\frac{1}{N}\overset{N}{\underset{i=1}{\sum }}x_{i}$	skew	$X_{ske}=\frac{\sum_{i=1}^{N}{\left(x_{i}-\bar{X}\right)}^{3}}{(N-1)X_{\sigma }{}^{3}}$
standard deviation	$X_{\sigma }=\sqrt{\frac{1}{N-1}\overset{N}{\underset{i=1}{\sum }}({x_{i}-\bar{X})}^{2}}$	kurtosis	$X_{kur}=\frac{\sum_{i=1}^{N}{\left(x_{i}-\bar{X}\right)}^{4}}{\left(N-1\right)X_{\sigma }{}^{4}}$
square root amplitude	$X_{r}={\left[\frac{1}{N}\overset{N}{\underset{i=1}{\sum }}\sqrt{\left|x_{i}\right|}\right]}^{2}$	crest	$I_{p}=\frac{X_{max}}{X_{rms}}$
absolute average amplitude	$\bar{X_{p}}=\frac{1}{N}\overset{N}{\underset{i=1}{\sum }}\left|x_{i}\right|$	margin	$I_{m}=\frac{X_{max}}{X_{r}}$
RMS	$X_{rms}=\sqrt{\frac{1}{N}\overset{N}{\underset{i=1}{\sum }}x_{i}{}^{2}}$	impulse	$I_{i}=\frac{X_{max}}{{\bar{X}}_{p}}$
peak to peak	$X_{p-p}=\max\left(x_{i}\right)-\min.\left(x_{i}\right)$	waveform	$I_{w}=\frac{X_{rms}}{\bar{X_{p}}}$

**Table 2 entropy-25-01539-t002:** IEEE PHM2012 bearing dataset information.

	Condition 1	Condition 2	Condition 3
Load(N)	4000	4200	5000
Speed(rpm)	1800	1650	1500
Training dataset	Bearing1-1	Bearing2-1	Bearing3-1
	Bearing1-2	Bearing2-2	Bearing3-2
Testing dataset	Bearing1-3	Bearing2-3	Bearing3-3
	Bearing1-4	Bearing2-4	
	Bearing1-5	Bearing2-5	
	Bearing1-6	Bearing2-6	
	Bearing1-7	Bearing2-7	

**Table 3 entropy-25-01539-t003:** IMS bearing dataset information.

Condition	Test 1	Test 2	Test 3
Bearing	T1-B1	T2-B1	T4-B1
	T1-B2	T2-B2	T4-B2
	T1-B3	T2-B3	T4-B3
	T1-B4	T2-B4	T4-B4

**Table 4 entropy-25-01539-t004:** Comparison of health indicators’ evaluations.

Bearing	Proposed Method-HI			RMS			ELM-AE-HI			SDAE-SOM-HI		
Criteria	Corr	Mon	Dis	Corr	Mon	Dis	Corr	Mon	Dis	Corr	Mon	Dis
Bearing1-3	0.78	0.16	0.14	0.77	0.14	0.15	0.77	0.14	0.44	0.67	0.12	0.62
Bearing1-4	0.33	0.17	0.26	0.32	0.16	0.33	0.31	0.15	0.70	0.41	0.17	0.76
Bearing1-5	0.29	0.13	0.20	0.16	0.12	0.21	0.28	0.12	0.63	0.26	0.14	0.55
Bearing1-6	0.25	0.11	0.13	0.10	0.10	0.12	0.18	0.11	0.86	0.31	0.13	0.79
Bearing1-7	0.37	0.14	0.22	0.23	0.11	0.26	0.34	0.10	0.77	0.35	0.14	0.58

**Table 5 entropy-25-01539-t005:** Determining the time to first prediction.

	Proposed Method	RRMS [5]	MSPC [32]	RMS [28]
Bearing1-1	27,210 s	25,720 s	11,290 s	14,620 s
Bearing1-2	8030 s	280 s	7620 s	8260 s
Bearing1-3	13,310 s	1140 s	8910 s	13,650 s
Bearing1-4	10,850 s	11,040 s	10,830 s	10,840 s

**Table 6 entropy-25-01539-t006:** Score of indicators for testing bearing performance.

Metric	Case	Paris Model	Degradation Trajectory Racking Model	Proposed Model
CRA	Bearing1-1	0.6967	0.7111	0.7243
	Bearing1-3	0.6317	0.5420	0.6521
	Bearing1-4	0.7443	0.7463	0.7483
Convergence	Bearing1-1	9234	9382	9187
	Bearing1-3	307.3	329.9	316.4
	Bearing1-4	315.3	296.7	292.1

## Data Availability

Not applicable.

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
