# Peer review of "A Multi-Featured Factor Analysis and Dynamic Window Rectification Method for Remaining Useful Life Prognosis of Rolling Bearings"

_entropy, 2023, doi:10.3390/e25111539_

Round 1

Reviewer 1 Report

Comments and Suggestions for Authors

A multi-featured Factor Analysis and Dynamic Window Rectification Method for Remaining Useful Life Prognosis of Rolling Bearings

Comments:

  1. The authors should add a nomenclature to identify all the parameters and abbreviations used through out the paper.
  2. of the notable strengths of this paper is its excellent use of subsections. The article is divided into clear and coherent sections, each with a specific focus, which greatly enhances the overall structure and readability. This organization facilitates easy navigation through the content and helps the reader to follow the logical progression of the study.
  3. In the introduction section of the manuscript the researchers did well briefly discussing the implication for industry. This takes the work from being a theoretical study to having practical and actionable insights. Further discussion on this in the discussion section could have tied up the paper well.
  4. In the introduction section, information could be organized in a logical and coherent manner to avoid hindering the reader’s understanding. The flow of the paper from the discussion of ball bearings to that of remaining useful life could be done better. Overall, the introduction is strong in content but needs restructuring to make it more comprehensible to readers.
  5. Figure 1 is well discussed in the introduction section, but the figure is found at a considerable distance in the methodology section. This spatial separation makes it challenging for readers to associate the degradation trajectory graph with the explanation of important points provided earlier in the introduction. The same issue is found with figure 3. Placing figures 1 and 3 in proximity with relevant discussions is recommended.
  6. The discussion for figure 3 could be discussed further, explaining each window and what range each window in the figure represents and the time value given as the accelerated degradation point unit should be written as 6780s and 7220s for figure 3a and b respectively.
  7. The authors do not validate or provide information on some mathematical variables mentioned in this manuscript, and this is a gap in the manuscript. For example, the parameters used in determining the RUL Prediction in section 2.5 should be indicated to know what each value of r,s,g stands for in the equation.
  8. Figure 6 & 7 show the relationship between the features, number of factors adn the variance of the common factor. The author of this manuscript should stay consistent to the associated naming of the number of factors as figure 6 gives a factor naming of 0-2 and figure 7, 1-3.
  9. Throughout the paper, redundancies can be found, which affects the reader’s clarity and conciseness of the paper. For example, the end of the introduction section and the beginning of the methodology section both give a summarized view on the procedure used to eliminate stochastic fluctuations in health indicators.
  10. The Author needs to indicate what the proposed method 5, 30 and 27 indicate in table 5 in the result section as the elbow point detection is a key determining section of the manuscript.
  11.  The discussion and summary section offers a comprehensive explanation of the work done and further work that needs to be done to generalize the method to other rolling bearing datasets.

Overall, the manuscript offers a solid statistical approach to different factors to monitor the remaining useful life of rolling bearings. However, it could benefit from having a simplified math indicating all its parameters and the discussion section could be more detailed sighting samples for how this method could be used in real-world applications. Adding insights on how this work fits within existing research could also enhance its impact. The paper needs refinement before it is considered for publication.

Author Response

Thank you very much for  your valuable comments, we have carefully revised the manuscript according to your adivce. please see the attachment.

Reviewer 2 Report

Comments and Suggestions for Authors

1. In the introduction, RMS is mentioned. For the first time, please provide the full name. Please review similar content throughout the article.

 2. At the end of the introduction, please explain the methods used in the article and the overall process of life prediction.

 3. Figure 2 depicts the entire life prediction process, which appears relatively simple. Please add a separate chapter at the end of Chapter 2 to describe the model in more detail.

4. It is best to enlarge the details of the two images in Figure 3 separately to make the images clearer.

 5. Please describe the final summary and conclusion separately.

Comments on the Quality of English Language

The quality of the English language is basically normal.

Author Response

(The authors gave the same response as above.)

Reviewer 3 Report

Comments and Suggestions for Authors

The research paper addresses the challenges associated with constructing health indicators (HI) and making the first prediction for the remaining useful life (RUL) of rolling bearings. The paper proposes a method based on multi-feature factor analysis; it begins by mining hidden common factors from multiple features using factor analysis. Subsequently, health indicators are constructed by maximizing the variance contribution following rotation. At this point, the Authors utilize a regression-based adaptive prediction model to understand the evolution trend of the health indicators and estimate the RUL of the bearings.

Importantly, the authors present experimental results that showcase the effectiveness of the health indicators in characterizing the degradation trend of bearings.

The content is, overall, interesting for the community of Structural Health Monitoring and Condition Monitoring experts. Nevertheless, some aspects should be improved before full acceptance. In more detail:

1.      While the abstract is generally clear in outlining the research methodology, it could benefit from providing more context regarding the significance of this research and its contribution to the existing body of knowledge.

2.      The abstract mentions the concept of a ‘cumulative gradient change’ in the trajectory of health indicators to determine the first prediction time; yet, this term does not seem to occur anymore in the text of the paper.

3.      The state of the art in Condition Monitoring for bearing faults could and should be expanded. Some recent works in the field include, e.g. https://doi.org/10.3390/app12031059.

4.      The paper includes the design of a dynamic window rectification method, which applies a sliding window-based gradient continuous change approach, to mitigate the effects of stochastic fluctuations in the data. However, it is not clear why an already-existing approach has not been used instead.

5.      Due to the specific journal, the implications of entropy in the context of the proposed procedure should be better highlighted.

6.      The authors acknowledge certain limitations in their approach. First, the method is most suitable for degradation processes with both slow and accelerated stages, characterized by elbow points. Additionally, the accuracy of the predictions depends on the quality of the constructed health indicators. Finally, the research has been validated on just two bearing datasets, suggesting the need for broader validation to ensure the method's generalizability. Some perspective future research lines are reported, however, it is not totally clear how these aspects could be eventually being dealt with.

7. Related to the above point, while the discussion section effectively summarizes the key findings and limitations of the research, it could benefit from further elaboration on the implications of these limitations, especially regarding scenarios without clear elbow points and the consequences of poorly constructed health indicators. Additionally, providing more specific details about the datasets and potential sources of bias would be valuable.

Comments on the Quality of English Language

The text is overall well-written and free from obvious grammatical or syntax issues, except for several missing blank spaces between words.

Author Response

(The authors gave the same response as above.)

Reviewer 4 Report

Comments and Suggestions for Authors

This paper is innovative and well-validated experimentally, but there are some problems. The reviewer concludes that the paper can be accepted for publication after revisions. There are some detailed comments and suggestions:

(1) In the introduction section, the manuscript does not provide an adequate overview of existing methods, which makes the article unconvincing and it is recommended that the authors add more state-of-the-art methods in recent years and point out the advantages of the proposed methods.

(2) The figures in the article should have been more elaborate; some of the figures in the article were so blurry that they were difficult to see and detracted from understanding the intent of the article

(3) The authors could consider investigating and introducing intelligent RUL prediction and fault diagnosis methods. The following literature may be helpful:

[1] A Recurrent Neural Network Based Health Indicator for Remaining Useful Life Prediction of Bearings. 10.1016/j.neucom.2017.02.045

[2] Lightweight Multiscale Convolutional Networks With Adaptive Pruning for Intelligent Fault Diagnosis of Train Bogie   Bearings in Edge Computing Scenarios. 10.1109/TIM.2022.3231325

Author Response

(The authors gave the same response as above.)

Round 2

Reviewer 1 Report

Comments and Suggestions for Authors

The authors have complied with the recommendations suggested.

The manuscript is ready for publication.

Reviewer 3 Report

Comments and Suggestions for Authors

This Reviewer is satisfied with the changes made and the reply to the first round of comments.

As a minor observation, the newly added lines 526-525 should not be double-spaced. Also, some further grammar checks can be useful at the proofreading stage.

Despite these editorial issues, however, the content of the paper is scientifically sound and the manuscript can be accepted after these minor revisions.

Comments on the Quality of English Language

The English of the paper has been improved.

Reviewer 4 Report

Comments and Suggestions for Authors

The authors have addressed all comments and suggestions. Thanks!